# Pre-clinical studies of *Schistosoma mansoni* vaccines: A scoping review

Emma L. Houlder[1]*, Lucas Ferreira da Silva[2], Angela van Diepen[1], Murilo Sena Amaral[3], R. Alan Wilson[4], Cornelis H. Hokke[1], Meta Roestenberg[1], Wilfried A.M. Bakker[5]

**1** Leiden University Center for Infectious Diseases, Leiden University Medical Center, Leiden, The Netherlands, **2** Harvard Medical School, Boston, Massachusetts, United States of America, **3** Laboratório de Ciclo Celular, Instituto Butantan, São Paulo, Brazil, **4** York Biomedical Research Institute, Department of Biology, University of York, York, United Kingdom, **5** Batavia Biosciences B.V., Bioscience Park Leiden, Zernikedreef 16, Leiden, The Netherlands

* e.l.houlder@lumc.nl

## Abstract

### Background

Schistosomiasis is caused by infection with worms of the genus *Schistosoma* including *S. mansoni*. Over 200 million people are infected, sterile immunity does not naturally develop, and no vaccine is available. This could be a critical tool to achieve control and elimination. Numerous candidates have been tested in pre-clinical models, but there is not yet an approved vaccine.

### Methodology/Principal Findings

We conducted a scoping review using a keyword search on Web of Science and a MeSH term search on PubMed. Articles were screened and included if they tested a defined vaccine candidate in a pre-clinical protection assay against *S. mansoni* between 1994–2024. Vaccine formulation, study design, and efficacy parameters from all articles were extracted. This data was summarised graphically, with the influence of different parameters appraised. A total of 141 candidate antigens were tested in 108 articles over the last 30 years, with most antigens tested only once and three (Sm-CatB, Sm-p80, and Sm-14) tested over 20 times. The median protective efficacy against worms was 35%. 10 antigens achieved over 60% efficacy, and only two (Sm-p80 and Sm-CatB) over 90%. Large variations in efficacy were observed with all repeatedly tested antigens, likely attributable to differing formulations and study designs. The effect of these varying parameters on the resultant efficacy was evaluated.

### Conclusions

A few vaccine candidates have achieved promising efficacy in pre-clinical studies. Most vaccines tested however have efficacy that falls short of that required for an impactful schistosomiasis vaccine. The diversity in study designs makes comparing

**Data availability statement:** Underlying data is included in the manuscript (S1 Table).

**Funding:** This work was supported by the European Union's Horizon 2020 research and innovation programme under the Marie Skłodowska-Curie grant (101063914 to ELH). The work was also supported by funding from the European Union (101080784 - WORMVACS2.0 to CHH, AvD and MR). The work was also supported by Conselho Nacional de Desenvolvimento Científico e Tecnológico (CNPq) (409278/2023-8 to MSA) and by Fundação de Amparo à Pesquisa do Estado de São Paulo (23/15324-8 to MSA). Views and opinions expressed are, however, those of the author(s) only and do not necessarily reflect those of the European Union or the European Health and Digital Executive Agency (HADEA). Neither the European Union nor the granting authority (HADEA) can be held responsible for them. Batavia Biosciences did not receive external funding for this research. The funders had no role in study design, data collection and analysis, decision to publish, or preparation of the manuscript.

**Competing interests:** The authors have declared that no competing interests exist.

vaccine targets a challenge. Use of consistent and optimized vaccine formulation (including adjuvant and platform) and study design parameters is critical to expedite the development of a schistosome vaccine.

---

## Author summary

Schistosomiasis, a major neglected tropical disease, is caused by infection with parasitic worms of the *Schistosoma* species, including *Schistosoma mansoni*. Individuals can be repeatedly re-infected, and there is no available vaccine. An initial stage of vaccine development is testing in a pre-clinical animal model. Here we have summarised tests of *Schistosoma mansoni* vaccines in the last three decades. 100+ vaccine candidates have been tested, with only 10 of these achieving efficacy of over 60%, and only 2 at over 90%. When the same vaccine candidate (antigen) is tested in a different formulation, or using different study design, the efficacy varies greatly. We have summarised the formulations and study designs used, and highlighted how certain parameters affect efficacy. Finally, we have assembled a series of recommendations to researchers on how to perform vaccine tests in the future.

## Introduction

Over 200 million people have schistosomiasis, with 264.3 million individuals requiring treatment in 2022 [1,2]. Most cases are in Africa, caused by infection with *S. mansoni* or *S. haematobium*, with *S. japonicum* causing the majority of Asian schistosomiasis. Approximately 10% of infected individuals develop severe disease, leading to 200,000 deaths per year [3,4]. Morbidity depends on the species, with intestinal/hepatic disease induced by *S. mansoni* and *S. japonicum*, and *S. haematobium* causing urogenital disease [5].

Schistosomiasis control relies upon a single chemotherapeutic (praziquantel), as well as water, sanitation and hygiene (WASH) interventions to reduce prevalence [6]. Praziquantel is effective at reducing worm burden. However, after treatment individuals can become rapidly re-infected upon exposure [6–8] and cases of parasite tolerance to the drug have been reported [9]. Whilst epidemiological evidence suggests that partial immunity can develop, with infection burden decreasing after teenage years, chronic morbidity including liver (periportal) fibrosis increases linearly to middle age [10–12]. A vaccine is urgently needed to stop infections and, therefore, the development of severe chronic morbidities. Between 2007–2022 schistosomiasis vaccine research was awarded 77 million USD of funding, the vast majority from public or philanthropic sources [13]. This amount of funding pales in comparison to other conditions, such as malaria (2576 million USD in the same period) [13]. The lack of commercial interest and high-level funding in schistosome vaccines makes the research highly vulnerable to funding gaps and dependent on personal or institutional interests, reducing our ability to perform structured assessments of vaccine candidates [14].

Induction of experimental protection against schistosome infection is possible, as demonstrated by the irradiated cercariae vaccine in animal models [15,16]. Exposure to large numbers of irradiated cercariae induces protective immune responses, which are thought to work via halting schistosome lung migration and therefore maturation [17]. Much effort has been put into replicating this protective effect with defined vaccine antigens [18,19]. A parallel line of research has sought to understand factors that are responsible for the (partial) protective immunity observed in endemic settings [20]. From these approaches, four defined vaccine candidates have progressed to human clinical trials – SchistoShield (Sm-p80, phase I), Sm-14 (phase II), The Human Schistosomiasis Vaccine (Sm-TSP2, phase II), and Bilhvax (Sh-28GST, phase III) [21–25]. The only vaccine that has yet progressed to phase III trials, Bilhvax (Sh-28GST), a vaccine for urogenital schistosomiasis, was not sufficiently efficacious [25].

Rationale: To test vaccine candidates, pre-clinical animal models are used. The model organism (usually a mouse) is first immunised and then challenged with a defined dose of schistosome cercariae. Vaccine efficacy is most commonly defined by a reduction of worm and/or egg counts, in comparison to an unvaccinated control group. Whilst several informative reviews on specific schistosomiasis vaccine candidates exist [14,18,19,26,27], pre-clinical vaccine research has not been systematically reviewed, making it difficult to understand the current state of the field.

Objectives: We aim to give an overview of pre-clinical prophylactic vaccine studies targeting *S. mansoni* in the last 30 years (1994–2024). We aim to summarise the overall landscape in terms of antigens tested, and their overall efficacy. Moreover, we focus on the parameters of a pre-clinical vaccination trial, for instance, the adjuvant and platform used, and the timing of immunization and challenge. At the same time, we explored how the choice of these parameters may influence the efficacy reached in these trials and discussed the translatability to clinical vaccination.

## Methods

### Literature search

We performed a literature search in June 2024 (detailed in Fig 1), to identify all schistosome pre-clinical vaccine studies. No review protocol exists. The initial search was performed on 25th June 2024 on Web of Science (all databases) using Title: (((schisto*) OR (mansoni) OR (haematobium) OR (japonicum)) AND ((protect*) OR (immunity) OR (immunis*) OR (immuniz*) OR (vaccin*))). News, awarded grants, dissertations, meeting abstracts, abstracts and retracted articles were not included in the search. This initial search resulted in 1622 articles. Two rounds of screening were then performed, to select articles that tested defined schistosome vaccine candidates in a *S. mansoni* protection assay using a model organism (Fig 1a). The initial screening was based on the title and abstract and resulted in the removal of 1342 articles: n = 166 reviews; n = 678 published before 1994; n = 242 not *S.mansoni*; and n = 266 not testing vaccines. The second round was more detailed, with the full text scanned and 175 articles excluded for the following reasons: n = 39 with no vaccine efficacy (e.g., immunogenicity or safety studies); n = 17 therapeutic vaccines or vaccines that prevent re-infection; n = 118 multiple antigens (e.g., bulk antigen, whole parasite, multi-protein constructs); and 1 article where the full text was not available. In addition, a supplementary search on PMID was performed, using the Mesh terms ("Schistosoma mansoni"[Mesh]) AND "Vaccines"[Mesh]). Papers found uniquely in the supplementary search were screened as before, leading to the inclusion of 3 additional articles. At the end of this process, 108 original articles remained (Fig 1). The Preferred Reporting Items for Systematic Reviews and Meta-Analyses extension for Scoping Reviews (PRISMA-ScR) checklist (S1 File PRISMA checklist), was used.

In addition, the following tests (experiments within an article) were excluded from the analysis: tests combining candidates; tests on therapeutic efficacy, or efficacy against previously *S. mansoni* exposed organism; tests in which the vaccine candidate has been mutated or inactivated [28–30]. Crosnier *et al.* 2019 [31] tested a large number [96] of vaccine antigens in the same formulation and study design, taking only egg counts for efficacy. To prevent this work from dominating formulation and study design figures (Figs 3–4) it has been excluded from these, with plots including this work shown in S2 Fig.

To understand the parameters that determine vaccine efficacy, data was charted by transferring information from these papers into a spreadsheet, with each row an individual test of a vaccine candidate. A total of 241 vaccine tests were recorded (S1 Table). Inputs in this spreadsheet were checked in a second round, by a single investigator. A single paper often contained multiple tests, for instance, if a candidate was tested with multiple adjuvants or in different mouse strains. For articles that reported the sequential testing of multiple partial antigens (such as epitopes or selected protein domains) from a single candidate, only the top-performing partial antigen was included in the analysis.

The same vaccine candidate may be referred to by several different names in prior literature. To account for this, the gene ID (Smp_XXXXXX) was therefore used, with each candidate also given one consistent short name. For some candidates, this gene ID was stated in the publication. If the gene ID was not stated, the candidate was mapped to a gene ID using (partial or complete) sequences referred to in the articles, using the WormBase ParaSite blast tool [31,32].

## Assessment of vaccine efficacy

Reported vaccine efficacy was recorded for all publications. To improve consistency between publications, efficacy was re-calculated using worm or egg counts reported in the paper. An appropriate adjuvant control was preferentially chosen as the control group for efficacy calculation, with an 'empty' (PBS/vehicle) control used if this was not possible. When the reported efficacy differed substantially (more than 10%) from the re-calculated efficacy, the re-calculated efficacy was used. Differences between re-calculated and reported efficacy were most often attributable to the use of the 'empty' (PBS/vehicle) control to calculate efficacy in the original paper, the use of an alternative efficacy formula, or the practice of reporting 0% efficacy when the vaccine increased worm/egg burden. Re-calculation of reported efficacy used the following formula:

$$100 \times \frac{\text{worm or egg count in control group} - \text{worm or egg count in vaccinated group}}{\text{worm or egg count in control group}}.$$

For egg efficacy analysis liver counts (and associated efficacy values) alone were used if possible. For a minority of tests (n = 4 [33,34]) only combined liver and intestine counts or just intestine counts (n = 4 [35,36]) were available, which were used in place of liver counts.

## Statistical analysis

To understand the effect of varying vaccine parameters on efficacy (worm reduction), a linear model was performed (ordinary least square). The following categorical parameters were included and the baselines used for each are as follows: gene ID - baseline "Smp_095360" (Sm-14); construct - baseline "full"; platform group - baseline "protein based", route of administration – baseline "skin". Adjuvant characteristics were included also as separate true/false parameters, with false as baseline - 'emulsion', alum', 'TLRagonist', 'cytokine', 'nanoparticle'. Tests were removed from the analysis if the Gene ID only appeared once in the database (e.g., the candidate has only been tested once). Finally the interval between challenge and cull and the number of immunizations were included as numeric variables. Since different features tested in our model could present some degree of correlation, we used the variance inflation factor (VIF) metric to detect and filter multicollinearity. The model was adjusted to reduce multicollinearity, and features such as "challenge dose" and "interval between immunization and challenge" were removed due to high VIF scores (over 5). This VIF threshold of five is conservative but is an important solution to remove multicollinearity as suggested by Akinwande *et al.* [37]. Only mouse studies were included in the modeling. In addition, a vaccine candidate (gene ID) was only included in the model if it had been tested more than once. To create the linear model and VIF filtering, we use the Python package `*stats models*` package (v0.14.4) [38].

## Data access

Data and graphs from this publication are also available on the web-based platform wormvaccines.nl, presented in an interactive and filterable format.

## Results

### Overview & efficacy

The review strategy is shown in Figs 1 and 2a. The quantity of research into defined schistosome vaccine antigens remained relatively consistent over the 30 years studied (1994–2024) (Fig 2b,c). The number of papers per year ranged between 1 and 10, with a median of 4 (Fig 2b). A total of 141 vaccine antigens were tested. The number of vaccine antigens tested per year ranged between 1 and 96 (outlier attributable to Crosnier et. al. [39]), with a median of 3 (Fig 2c).

Prophylactic efficacy (reducing future infection) is the key outcome by which vaccines are evaluated. The majority (58%) of articles measured efficacy by both worm and egg counts, with (38%) measuring only worm counts, and 3% measuring only egg counts (Fig 2d). There was a significant positive correlation between the efficacy observed between egg and worm counts (Fig 2e).

The median efficacy of all tests when determined via worm count was 35% with an interquartile range (IQR) of 18–50% (Fig 2f). The median efficacy when determined via egg count was 29%, with an IQR of 2–53% (Fig 2g). The majority of vaccine antigens have been tested only once, with a small number (n = 6) undergoing 10 or more tests, 3 of which (Sm-CatB, Sm-14, and Sm-p80) have been tested over 20 times since 1994. A few candidates have been tested independently in different research groups, particularly Sm-14 and Sm-28GST. The total number of model organisms used, number of tests and articles, independent testing, as well as the maximum efficacy for each antigen tested in more than one article is summarized in Table 1.

Focusing on either worm or egg efficacy against primary infection, in the past 30 years only two candidates (Sm-p80, Sm-CatB) have reached an efficacy of over 90% [62,77]. Ten candidates have achieved a maximum efficacy of over 60%. These include the three candidates currently in human clinical trials (Sm-14, Sm-p80, and Sm-TSP2) [44,59,62,111] as well as seven other defined schistosome candidates (Sm-28GST, Sm-CatB, Sm-GPX, Sm-CL3, Sm-CT-SOD, Sm-29 and Sm-KI-1) [29,30,35,70,72–75,77,78,89,99,119–121]. The efficacy of these candidates varies widely between individual tests (Fig 2h). This variation highlights the critical importance of other parameters in vaccine testing, including how the vaccine is produced (e.g., adjuvant, platform) or the study design (e.g., number of doses, model organism chosen).

### Vaccine formulation

**Platform.** Schistosome vaccines have been produced in several different platforms (Figs 3a and S1a). The most common of these are protein-based systems, mainly recombinant proteins but also native and peptide constructs. The next most common choice was DNA vaccine systems. Fewer antigens have also been tested in systems using recombinant microbes, such as vaccinia virus or *Salmonella*. A combination of DNA and protein vaccine approaches, as well as protein and recombinant microbe approaches in heterologous prime/boost systems, has been performed, although rarely (Fig 3a).

To understand if one type of platform is the most efficacious, two approaches were taken. First, the maximum efficacy against worms for each antigen in each platform group was plotted, which showed a trend for increased efficacy with recombinant microbe approaches (Fig 3b). This approach (Fig 3b) may be biased as recombinant microbe approaches have generally been used only with antigens with higher inherent efficacy. To account for this, Fig 3c shows the change in median efficacies of different platforms within the same antigen, relative to protein-based platforms. The data suggest a tendency for higher efficacy with protein-based approaches.

For protein production systems, how the protein or peptide is produced can be critical to determine crucial folding and posttranslational modifications [122,123]. Most antigens have been produced as full constructs (75%, Fig 3d), with

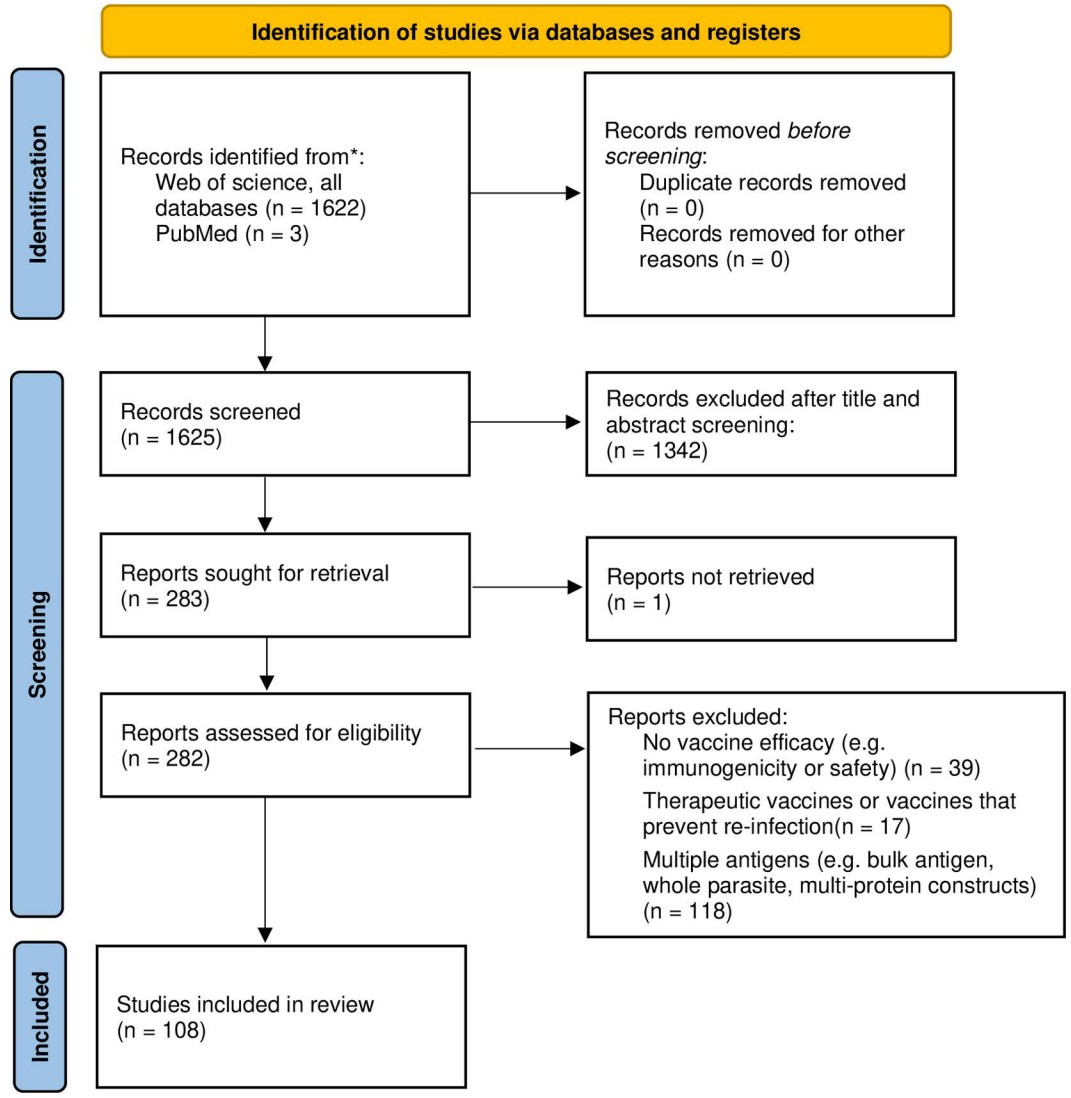

**Fig 1. PRISMA 2020 flow diagram including searches of databases and registers only.**

a smaller proportion also tested as partial constructs (e.g., peptides, subunits, or fragments). Within candidates, efficacy was consistently higher in full rather than partial construct approaches (Fig 3e). The majority of proteins tested in schistosome vaccine trials have been produced in *E. coli* systems (Fig 3f). Once a protein production system has been tested, this system tends to be used for the majority of tests of that antigen, with few comparisons of protein production systems within one antigen (S1b Fig).

**Adjuvant.** Adjuvants are the parameter most often changed in schistosome vaccine experiments. The most common formulation used is alum, followed by CFA/IFA (Fig 3g). Beyond these common formulations a wide variety of adjuvants have been used (Fig 3g). This is exemplified when focusing on the vaccine antigens tested over 10 times (Sm-28GST, Sm-CatB, Sm-14, Sm-23, Sm-29 and Sm-p80), which have each been tested with between 3–11 different adjuvants (S1c Fig). Adjuvants were categorized according to different characteristics, for example inclusion of an emulsion or alum element, with one adjuvant potentially containing multiple elements (Fig 3g). Adjuvants

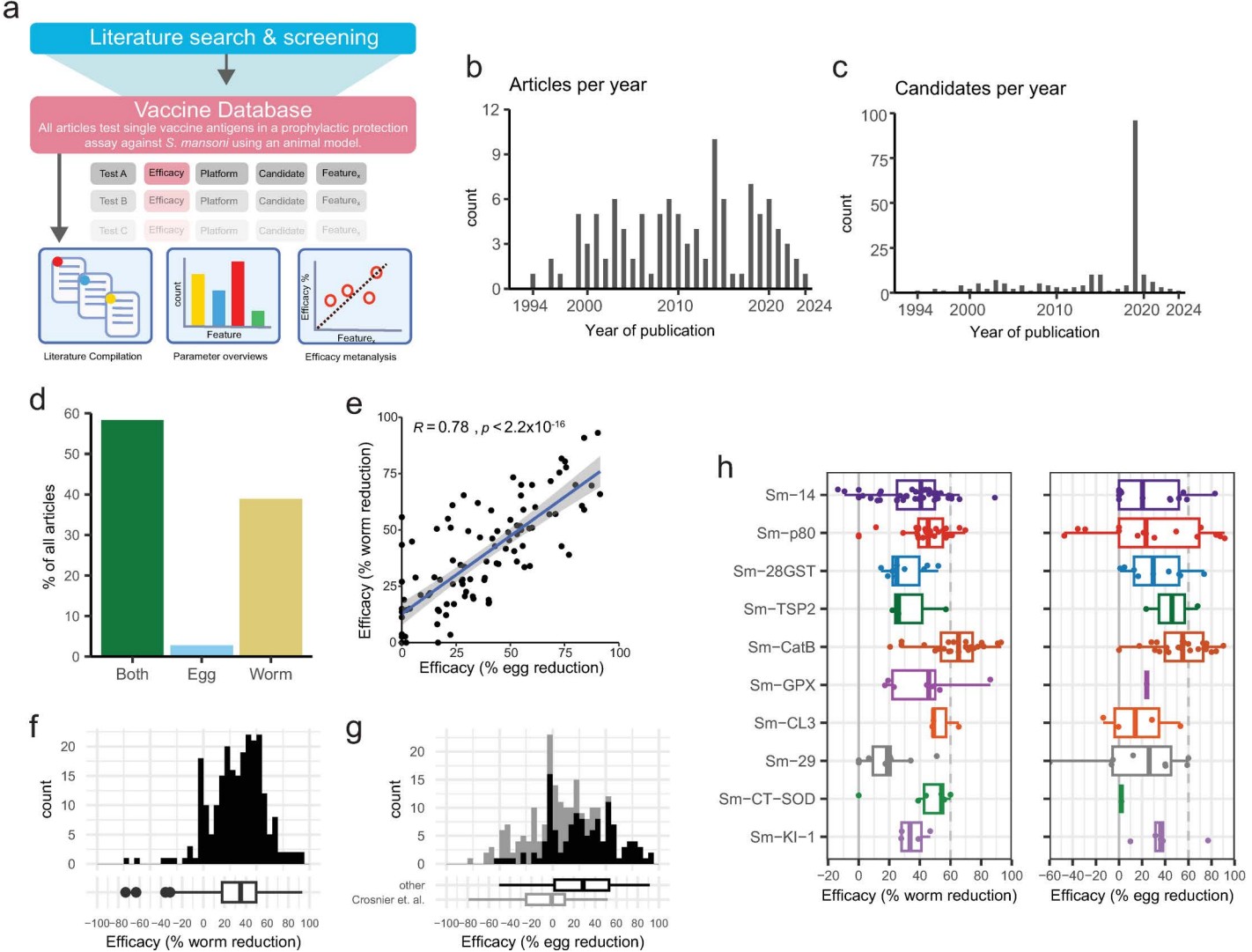

**Fig 2. Overview and efficacy.** a) Overview of review strategy b) Histogram of the number of included articles per year and c) candidate antigens tested per year between 1994 and 2024. d) Proportion of candidates where vaccine efficacy is tested via worm counts, egg counts or both. e) Correlation between worm and egg efficacy in all tests in which both were measured. The blue line and associated grey confidence interval represents a linear regression, with pearson's correlation coefficient and p value reported. f & g) Histogram of the efficacy against worms (f) or eggs (g) in all vaccine tests, summarised in boxplot below. h) Boxplots showing the efficacy against worms (left) or eggs (right) in all tests of antigens that have ever achieved over 60% efficacy.

applicable for human use highlighted (Fig 3g). There was no clear difference in efficacy between different adjuvant types when the maximum efficacy for all antigens using the different adjuvants was plotted (Fig 3h). However, when the change in median efficacy with different adjuvant types within one antigen (compared to no adjuvant) was plotted, the most consistent increase in efficacy was seen in adjuvants with an emulsion element (Fig 3i). The most efficacious adjuvants were Addavax (91% efficacy against worms) and montanide (76% efficacy against worms) in combination with Sm-CatB, the emulsion/TLR4 agonist GLA-SE (91% against eggs) or the TLR9 agonist CpG (87% efficacy against eggs) with Sm-p80 and the emulsion/TLR agonist CFA for Sm-14 (89% efficacy against worms) [44,59,62,70,72].

**Table 1. Maximum efficacy of each of the antigens tested in more than one article.**

| GeneID | Name | Animals used | Tests | Articles | Independent testing* | | % max efficacy (worm) | % max efficacy (egg) | References |
|---|---|---|---|---|---|---|---|---|---|
| | | | | | Last author | Any author | | | |
| Smp_095360 | Sm-14 | 712 | 45 | 17 | 12 | 6 | 89 | 83 | [36,40–55] |
| Smp_214190 | Sm-p80 | 404 | 30 | 16 | 5 | 2 | 70 | 91 | [28,34,56–69] |
| Smp_067060 | Sm-CatB | 227 | 24 | 11 | 5 | 3 | 93 | 90 | [30,39,70–78] |
| Smp_072190 | Sm-29 | 115 | 11 | 8 | 5 | 3 | 51 | 60 | [35,39,50,51,79–82] |
| Smp_333010 | Sm-28GST | 145 | 12 | 7 | 7 | 7 | 52 | 74 | [83–89] |
| Smp_017430 | Sm-23 | 97 | 12 | 4 | 2 | 2 | 45 | 39 | [90–93] |
| Smp_056970 | Sm-37GAPDH | 81 | 7 | 4 | 3 | 2 | 42 | 48 | [94–97] |
| Smp_311670 | Sm-KI-1 | 48 | 5 | 4 | 4 | 4 | 47 | 77 | [98–100] |
| Smp_003300 | Sm-9b | 120 | 9 | 3 | 2 | 1 | 55 | Not tested | [101–103] |
| Smp_017730 | Sm-200 | 27 | 4 | 3 | 3 | 3 | 38 | 0 | [39,104,105] |
| Smp_045200 | Sm-22.6 | 60 | 5 | 3 | 2 | 1 | 34 | 0 | [80,106,107] |
| Smp_058690 | Sm-GPX | 56 | 8 | 3 | 1 | 1 | 86 | 24 | [29,108,109] |
| Smp_176200 | Sm-CT-SOD | 63 | 8 | 3 | 1 | 1 | 60 | 2 | [29,108,109] |
| Smp_210500 | Sm-CL3 | 47 | 4 | 3 | 2 | 2 | 65 | 53 | [39,73,76] |
| Smp_335630 | Sm-TSP2 | 45 | 3 | 3 | 2 | 2 | 57 | 68 | [82,110,111] |
| Smp_075800 | Sm-32 | 17 | 2 | 2 | 2 | 2 | -4 | 13 | [39,112] |
| Smp_095980 | Sm-ECSOD | 24 | 3 | 2 | 1 | 1 | 40 | 33 | [29,108] |
| Smp_105450 | Sm-SLP1 | 14 | 2 | 2 | 2 | 2 | -36 | 23 | [39,113] |
| Smp_136690 | Sm-AChE2 | 15 | 2 | 2 | 2 | 2 | 35 | 42 | [39,114] |
| Smp_138060 | Sm-MEG3.2 | 14 | 2 | 2 | 2 | 2 | 8 | -16 | [39,115] |
| Smp_153390 | Sm-NPP-5 | 14 | 2 | 2 | 2 | 2 | 0 | 17 | [39,116] |
| Smp_330190 | Sm-Rho | 30 | 6 | 2 | 1 | 1 | 24 | 17 | [117,118] |

*Independent testing measures the number of independent research groups testing one antigen. Last author counts the number of unique last authors, Any author counts the number of articles with no overlapping authors.

## Study design

**Animal model.** Over 2800 individual animals have been used for schistosome vaccine studies in the last 30 years, the vast majority of which are mice, with a few studies performed in rats, baboons and rabbits (Fig 4a). Only four vaccine antigens (Sm-p80, Sm-CT-SOD, Sm-ECSOD and Sm-GPX) have been tested in baboon models in the last 30 years. Only Sm-p80 has been tested multiple times with different formulations and experimental designs in baboons [56,58,61,62,67,69,124,125]. Within mouse models, several different mouse strains have been used, with BALB/c and C57BL/6 being the most common (Figs 4b, and S1d).

**Immunization route, frequency and challenge timing.** Most vaccines are given in 3 doses (1 prime, 2 boost) prior to schistosome challenge (Figs 4c, and S1e). The majority of vaccines have been tested via the subcutaneous route, closely followed by intraperitoneal and intramuscular routes (Figs 4d and S1f). When efficacy was plotted in comparison to vaccination route, there tended to be an increase in efficacy when mixed routes were employed (Fig 4e).

The dose most commonly used for challenge is 100 +/- 25 cercariae (Figs 4f and S1g). When the maximum efficacy for each antigen at each challenge dose is plotted, excluding non-mouse studies which generally use a much higher challenge dose, the highest efficacy tended to be observed with a challenge of 150 +/- 25 cercariae (Fig 4g).

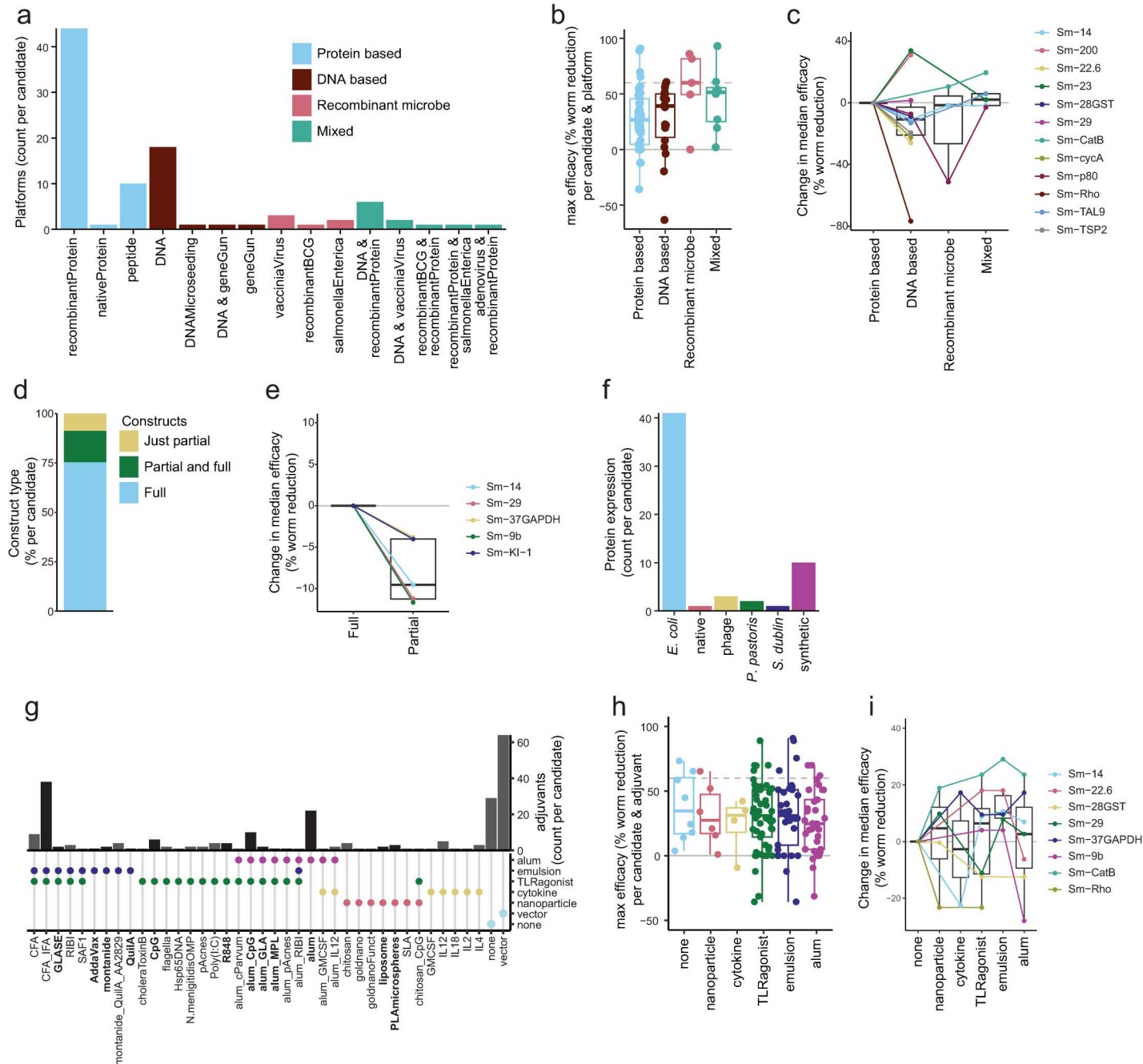

**Fig 3. Vaccine formulation.** a) Barplot showing platforms used for each schistosome vaccine antigen. b) Boxplot showing the maximum efficacy of each schistosome vaccine candidate by platform type tested. c) Line and boxplots showing change in median efficacy of candidates tested in different platform types, compared to protein based platforms. d) Proportion of candidates tested as full constructs, partial constructs or both. e) Line and boxplots showing change in median efficacy of candidates tested as partial constructs, compared to full constructs. f) Barplot showing recombinant protein system used for each protein based tested antigen. g) Bar and dotplot showing adjuvants used for each tested antigen, with adjuvant characteristics indicated on the dotplot below. Adjuvants applicable for human use (or with a similar formation to human-applicable adjuvants) are shown in bold, with black bars. h) Boxplot showing the maximum efficacy of each schistosome vaccine candidate by adjuvant characteristics. As adjuvants may have multiple characteristics, a single test of a vaccine antigen may appear in multiple boxplots. i) Line and boxplots showing change in median efficacy of candidates tested as proteins or peptides with different adjuvants, compared to no adjuvant. When count per candidate is plotted, each candidate is counted once per group (e.g., platform) to avoid over-representing frequently tested candidates and provide a balanced overview.

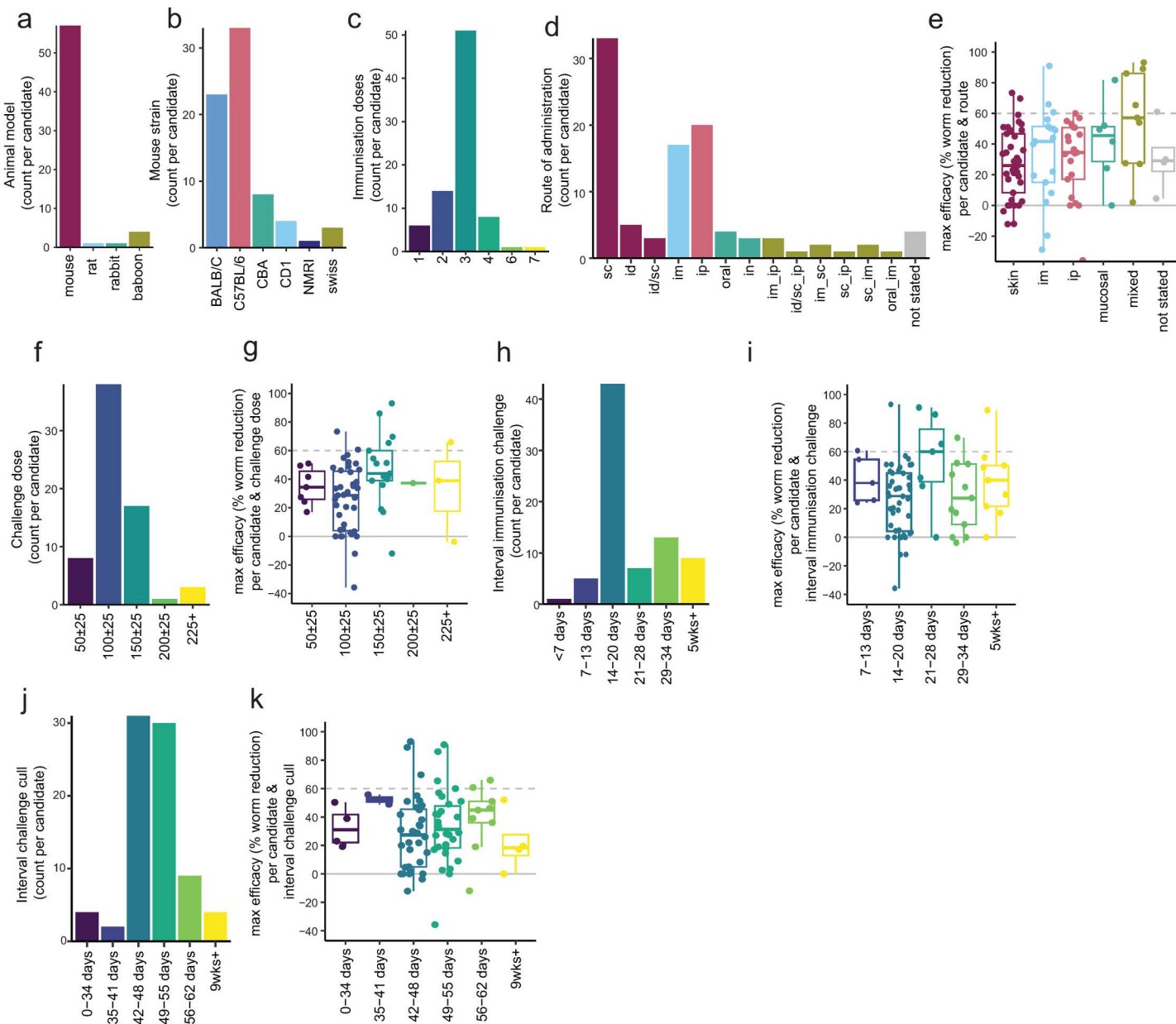

**Fig 4. Study design.** a) Barplot showing animal model used for each tested antigen. b) Barplot showing mouse strain used for each antigen tested in a mouse model. c) Barplot showing the number of immunisation doses used for each tested antigen. d) Barplot showing administration route for each vaccine antigen. e) Boxplot showing the maximum efficacy of each schistosome vaccine candidate by type of administration route. f) Barplot showing cercarial challenge dose for each vaccine antigen. g) Boxplot showing the maximum efficacy of each schistosome vaccine candidate, grouped by number of cercariae used in the challenge. h) Barplot showing gap between final immunisation and challenge for each vaccine antigen. i) Boxplot showing the maximum efficacy of each schistosome vaccine candidate by the interval between immunisation and challenge j) Barplot showing gap between cercariae challenge and cull (measurement of worm/egg counts) for each vaccine antigen. k) Boxplot showing the maximum efficacy of each schistosome vaccine candidate, grouped by the interval between challenge and cull (measurement of worm/egg counts). When count per candidate is plotted, each candidate is counted once per group (e.g., animal model) to avoid over-representing frequently tested candidates and provide a balanced overview.

The timing of the challenge after immunization as well as the interval between challenge and worm count could impact efficacy. The majority of tests have 14–20 days interval between last vaccination and challenge (Figs 4h and S1h). There is no clear effect on efficacy when looking at how each antigen performs at different gaps between immunization and challenge, although the highest efficacy tended to be observed at 21–28 days (Fig 4i). Another critical factor of timing is the gap between challenge and cull. The most common gap used between challenge and cull is 42–48 days (6 weeks) (Figs 4j and S1i). There was no clear relation between the gap between challenge and cull and the efficacy (Fig 4k).

**Linear model of vaccine efficacy.** To further understand how vaccine formulation and study design parameters affected vaccine efficacy a linear model was used (Fig 5). Only candidates tested more than once were included, and Sm-14 was used as the baseline. The candidates Sm-CatB and Sm-p80 had a significant positive effect on efficacy (worm reduction), whilst the candidates Sm-Rho and Sm-22.6 had a negative effect. In terms of study design parameters, use of a mixed administration route significantly increased vaccine efficacy (compared to baseline skin administration), whilst using a partial construct (e.g., subunit or peptide) decreased vaccine efficacy (compared to baseline full construct) (Fig 5). No significant effects on vaccine efficacy were found with the parameters platform group, adjuvant, the number of immunization doses and the interval between challenge and cull. The lack of significant effect for these parameters does not preclude them from having a role in vaccine efficacy. Parameters such as adjuvant:emulsion that had non-significant positive effects may reach significance with further pre-clinical testing.

## Discussion

### Overview & efficacy

There is not enough schistosome vaccine research, with only three vaccines in active clinical development for schistosomiasis, in comparison to 15 respiratory syncytial virus and 38 malaria vaccines in active clinical development (Phase I-IV) [126,127]. Whilst there is no commonly agreed upon target product profile, minimal efficacy against re-infection is proposed at 60% [128], with modelling studies to estimate the health impact and cost-effectiveness of schistosome vaccination commonly assuming efficacies of 75–100% (reviewed in [129]). A large gap therefore exists between the efficacy required to obtain health impact and that observed in most pre-clinical studies. Whilst ten defined schistosome antigens in the last 30 years have achieved higher than 60% efficacy in a single test, no candidate that has undergone repeated

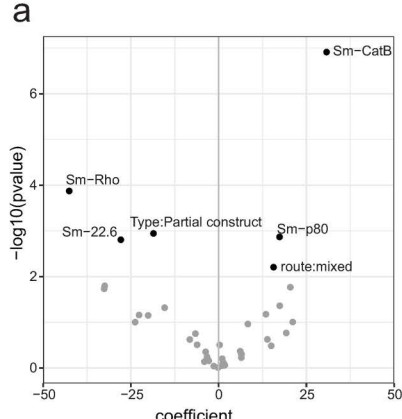
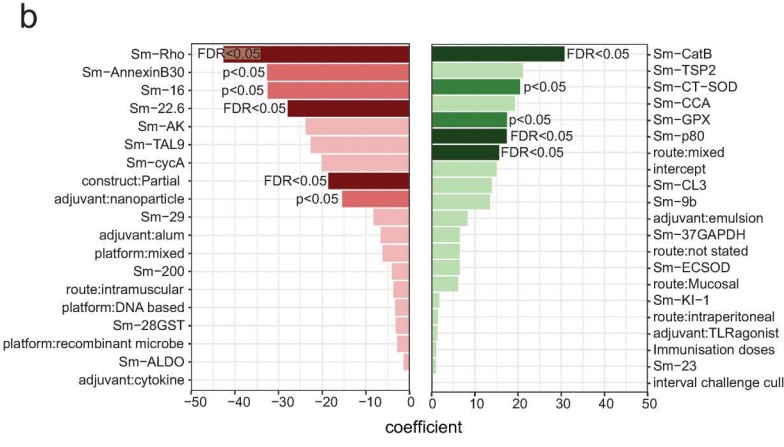

**Fig 5. Linear model of parameter effect on efficacy.** A linear model was fitted to dissect how the choice of candate, vaccine formulation and study design impacted efficacy (worm reduction). a) Volcano plot showing significant (FDR<0.05) parameters. b) Bar plot showing the coefficients for all variables included in the model. Red denotes a negative coefficient, green a positive coefficient. Variables that significantly affected the efficacy are shown in a mid colour (p<0.05) or dark colour (FDR<0.05).

testing has a median efficacy of higher than 60% for both worms and eggs. It is therefore clear that not only must the correct candidate be picked, but the vaccine formulation and study design must be optimized for efficacy.

The main outputs for schistosome vaccine studies are worm and liver egg counts. Whilst these measures are highly related, measuring both is recommended, as vaccines may show a specific anti-fecundity effect. If vaccination specifically leads to fewer (viable) eggs being laid, greater efficacy will be seen against eggs than worms, as demonstrated for Sm-p80, Sm-32 and Sm-KI [59,62,99]. For further depth, secondary outcomes measured could include egg viability and fecal egg counts, with unviable eggs less able to transmit disease [62,67,112,130]. In primate studies morbidity measures could also be taken – with one study (Sm-CT-SOD in baboons) finding only moderate parasitological efficacy but large reductions in intestinal symptoms (diarrhoea) [108]. When multiple outcomes are measured the primary outcome should be defined (and ideally registered) in advance to avoid misleading 'cherry picking' by publishing only the most efficacious outcome [131].

### Antigen choice

Diverse rationales are given for choosing antigens as schistosome vaccine candidates, and which is 'best' is still unclear. Our analysis using a linear model (Fig 5) has highlighted Sm-CatB as well as Sm-p80 as candidates that can significantly increase vaccine efficacy. Although non-significant, potentially due to reduced numbers of tests, the other candidates (including Sm-TSP2 and Sm-CL3) that had positive coefficients could warrant further study. In the context of a multicellular parasite, combinations or "cocktails" of candidates, which have shown efficacy in other helminth infections [132–134], may be required to combat functional redundancy of targets and increase vaccine efficacy [14,18,19,27,135].

### Platform

The platform of a vaccine partly determines the immune response induced, and therefore efficacy. In our analysis full constructs (Figs 3e and 5) and protein based platforms have emerged as the most efficacious, in general (Fig 3c) and when focusing on the specific 'front-runner' antigens such as Sm-p80 and Sm-14 [44,62]. This may be due to lower IgG levels induced in DNA vaccine approaches [107,136], with IgG proposed as a key correlate of protection [137,138]. All three vaccines currently in clinical development are recombinant proteins, with protein production technology scalable and cost-effective for clinical use [21–23,139].

The choice of a protein production system is critical for proper protein folding, targeting and post-translational modification (including glycosylation). These may be critical for the presentation of conformational epitopes as well as protein function, potentially required to induce a protective response [30]. Most antigens have been tested in only one production system, and therefore we have not been able to analyze this here. However, in situations where antigens have shown unusually low efficacy, for instance Sm-p80 in an RVV vector, or Sm-CatB produced in HEK293 cells, use of a non-optimal expression system could be considered as a potential explanation [28,39]. Immunogenicity and therefore efficacy could be increased with proper post-translational modification of a protein. This was recently shown, where increased efficacy of a parasitic roundworm antigen was seen when the antigen was expressed in a plant-based system, engineered to mimic glycosylation of the native antigen [122].

Prioritization of protein systems has the potential to overlook novel platforms which could be useful alone, or in combination with a protein platform [140]. Notably lacking is testing of mRNA vaccines, which have proven efficacy in the coronavirus pandemic [141], as well as virus-like-particles which have been used in the approved R21 malaria vaccine [142]. A particularly successful example of use of a novel platform is the *S. enterica* vector with Sm-CatB, which was able to induce over 80% protection alone, and over 90% when combined with the recombinant protein [74]. The increased efficacy of heterologous protein and *Salmonella* vectored vaccination could be attributable to increased breadth of the immune response, inducing both IgG and IgA antibodies [74]. Heterologous platform combinations have shown increased efficacy in some [74,75,77], but not all [29,40,42,90], schistosome vaccine studies.

## Adjuvant

Whilst a wide variety of adjuvants have been tested in schistosomiasis, comparisons are difficult due to the wide array of adjuvants being tested within different experimental conditions [143]. A key exception to this is a recent study that tested six adjuvants side-to-side in a Sm-CatB vaccination model, which found the highest efficacy with the adjuvant Addavax, followed by montanide, Alum/MPL, Alum/CpG, SLA and finally Alum [70].

In our analysis, adjuvants with an emulsion element (including CFA/IFA, Addavax, GLA-SE) have provided the most consistent increase in efficacy (Fig 3g) [61,70]. Of the antigens currently in clinical trials Sm-14 and Sm-p80 use the TLR4-ligand GLA in a stable emulsion (GLA-SE), whilst Sm-TSP2 uses GLA in an aqueous formation (GLA-AF) [21,22,24,139]. Translatability of pre-clinical studies should increase if human-applicable adjuvants are used, especially those available for global health purposes. To do this, the use of initiatives such as the Vaccine Formulation Institute or Access to Advanced Health Institute is encouraged.

## Animal models

The vast majority of pre-clinical research uses mice as the animal model. This has multiple benefits, with mice relatively small and cheap allowing for sufficient animals for well-powered studies. Moreover, the widespread use of mice allows comparability between studies and has meant reagents are available for immune monitoring and profiling. However, it is unclear how well murine findings can translate to humans. Translation from murine to non-human primates has shown mixed response, with Sm-p80 showing relatively consistent efficacy across baboon and mouse studies [56], whilst a separate study on antioxidant enzymes (Sm-CT-SOD, Sm-EC-SOD, Sm-GPX) showed reduced efficacy when tested in the baboon model [108]. Critically only a small proportion of penetrating cercariae (estimated at 32%) mature in the mice, likely due to the physiological challenge presented in the lung capillaries [144]. This makes mice particularly susceptible to 'spurious' protection data if the integrity of pulmonary capillaries is compromised by an ongoing innate inflammatory response, induced by an adjuvant or immunogenic protein [144].

## Immunization route, frequency and challenge timing

There are many factors underlying the choice of vaccination schedule. Administration of multiple doses is often required to reach protective antibody titers, with most investigators choosing 3 doses (Fig 4c). Parenteral administration of vaccines (subcutaneous, intradermal, and intramuscular) is common in pre-clinical testing and has been shown to induce a strong immune response, with intradermal administration often achieving a similar response with reduced dosage [145]. In our linear model we found that the use of mixed administration routes significantly increased vaccine efficacy (Fig 5). Notably, studies that used mixed administration routes often used different platforms, so it is difficult to unpick these two factors. The use of mixed vaccination routes may increase the strength and breadth of the immune response [146].

A recent review has highlighted the duration of protection as of equal importance to efficacy, stating that if protection is less than five years vaccination will have very little impact, with repeated vaccination needed [129,147,148]. Despite this, the longest gap between vaccination and challenge in any pre-clinical study included in our analysis was two months ([44], Sm-14), with the other antigens in active clinical development having been tested at most 15 (Sm-TSP2) and 28 days (Sm-p80) post-immunization [62,111]. Increasing the interval between immunization and challenge is particularly crucial, as discussed earlier, in the context of the mouse model, where innate inflammation and damage in the lung capillaries may lead to non-specific and spurious efficacy [144]. Encouragingly, increasing the interval between immunization and challenge did not reduce efficacy in the studies reported here (Fig 4i), with increased intervals reported to reduce variability in egg-counts and therefore statistical power [39].

Schistosome migration and maturation determine the optimal interval between cercarial challenge and taking parasitological outputs (worm/egg counts). Schistosome migration is not a straight line, with one to several circuits of the systemic and pulmonary vasculature before they arrive in the splanchnic arteries and liver. Egg production begins 5–6 weeks post infection

[4]. If parasitological outputs are taken too early then any vaccine-induced delay in the migration or maturation process may manifest as protection (particularly with egg counts), a finding that would disappear if these outputs had been taken later.

Natural infection with schistosomes is likely to occur via low-level 'trickle' exposure to genetically diverse cercariae, in stark contrast to the bolus exposure of lab-adapted schistosomes in pre-clinical or experimental vaccine trials. Robust immune responses to high burdens of challenge cercariae may synergize with vaccine responses, resulting in efficacy that would not be replicated in natural trickle exposure [149,150]. Encouragingly however, when a schistosome vaccine construct (Sm-37.5 and Sm-10-DLC) was tested in either a bolus (120 cercariae) or repeat (6 x20 cercariae) infection schedule similar efficacies (30% and 21% respectively) were observed [95]. Genetic diversity exists within *S. mansoni* field isolates, and whether a vaccine that is efficacious against lab-adapted *S. mansoni* strains will work against diverse endemic strains remains to be determined [151].

## Conclusions

There is wide variation in the efficacy of pre-clinical schistosome vaccines achieved using the same vaccine candidate in different formulations. All of the vaccines currently in clinical trials have shown poor efficacy (<30%) in individual studies, using sub-optimal vaccine formulations [28,47,82]. When lessons from pre-clinical studies are taken on to clinical translation, a good example is the evidence-based use of the GLA-SE emulsion adjuvant in the clinical formulation of Sm-p80 [62], the reduced efficacy of other formulations should not detract from the potential of the vaccine. It does however make it difficult to understand what is a 'promising' antigen that should be further optimized, and raises the question if other antigens have been prematurely ignored due to low efficacy found in sub-optimal formulations or study designs.

Using standardized parameters for initial testing of schistosome antigens would allow us to better compare between antigens and therefore more systematically prioritize which should be further tested. In Table 2, we have recommended these parameters, prioritizing choices that maximize comparability with prior studies, improve efficacy, and/or best translate to clinical use. Whilst maximizing comparability is crucial, these recommendations are not meant to stifle innovation - changing a limited number of parameters (for instance to allow testing of a novel adjuvant or platform) is encouraged.

**Table 2. Recommended parameters for initial preclinical testing based on current literature. Comparability to prior literature, impacts on vaccine efficacy and translatability are summarised. The factor driving the recommendation is shown in bold.**

| Parameter | Recommendation | Comparability | Efficacy | Translatability |
|---|---|---|---|---|
| **Platform/Construct** | **Full recombinant protein** | Commonly used (Fig 3a & 3d) | **Increase (Fig 3c & 3e, 5)** | Easy to scale up for clinical use. |
| **Adjuvant** | **Emulsion based (+/- TLR agonist)** | Commonly used (Fig 3g) | **Increase (Fig 3i)** | Emulsion adjuvants such as GLA-SE and SWE suitable for human use and available for global health applications. |
| **Animal model** | **Mouse (Balb/c or C57BL/6)** | **Commonly used (Fig 4a&4b)** | unknown | Translatability to humans unclear. |
| **Number of immunisations** | **Three doses** | **Commonly used (Fig 4c)** | unknown | Three dose administration fit into standard childhood vaccination schemes. |
| **Route of immunisation** | **Parenteral (s.c./i.d./i.m.)** | Commonly used (Fig 4d) | No effect (Fig 4e) | **Parenteral administration most-used human vaccination route.** |
| **Cercarial challenge dose** | **100 cercariae** | **Commonly used (Fig 4f)** | No effect (Fig 4g) | Lower doses reflective of natural trickle exposure, but introduce greater variability in worm/egg counts, 100 cercariae is a compromise between these factors. |
| **Interval immunisation and challenge** | **At least 2 weeks, preferably 3–4.** | Commonly used (Fig 4h) | No effect (Fig 4i) | **Durable protection required for vaccine to have real-world disease impact and cost-effectiveness.** |
| **Interval challenge and worm/egg counts** | **Seven weeks** | Commonly used (Fig 4j) | No effect (Fig 4k) | **Shorter intervals may lead to anomalous 'efficacy' due to delayed schistosome migration, which would not translate to efficacy seen in long-term clinical trials.** |

After achieving promising results in initial testing, further innovation can be performed to increase efficacy and translatability. Innovative platforms (mRNA, viral vectors, etc) could be tested alone or in combination, antigens could be combined into cocktail vaccinations, and recombinant protein production systems could be optimized for immunogenicity [122,152]. For a schistosome vaccine to show clinical impact durability of protection is almost as critical as efficacy, and therefore long-term (month to years) rechallenges post-vaccination are crucial in the pre-clinical pipeline [129]. Determination of IgE reactivity in endemic areas is crucial to prevent hypersensitivity responses to immunization [153]. Definition of a target product profile or preferred product characteristics, as available for other conditions, is sorely needed to guide this later-stage development [128,154].

There are several limitations to this analysis. The review does not discuss correlates of protection, which have been recently reviewed elsewhere [15]. The focus on a 30-year time period means this review does not provide an overview of all vaccine tests prior to 1994. For practical reasons, the efficacy data used in the analysis does not consider statistical significance; this may over-emphasize the efficacy of antigens from studies with high variability using low numbers of model organisms. The review is solely focusing on *S. mansoni*, and therefore limited applicability to other schistosome species. A crucial limitation is that it was only possible to review published studies, with negative (unprotective) results often left unpublished.

Rational design of pre-clinical studies is required to expedite the progress of schistosome vaccines into clinical trials and real-world use. To do this we must learn from the decades of schistosome research that have been previously undertaken. A key next step is understanding how well pre-clinical models translate to clinical efficacy. Controlled human infection models are a cost-effective option to test multiple vaccine candidates and can guide traditional Phase III efficacy studies as well as back-translate to prioritize the antigens and formulations to test in pre-clinical models [155,156].

## Supporting information

**S1 File. PRISMA checklist.** Checklist of PRISMA guidelines for scoping reviews.
(PDF)

**S1 Fig. Vaccine formulation and study design parameter variation across 6 most tested vaccine candidates.** Stacked barplots show the number of tests with the indicated parameter a) platforms, b) recombinant protein production, c) adjuvants, d) mouse strain, e) immunisation doses, f) administration route, g) challenge dose, h) gap immunisation and challenge, i) gap challenge and cull.
(PDF)

**S2 Fig. Vaccine formulation and study design barplots including *Crosnier et al.* 2019.** a) Platforms used for each schistosome vaccine antigen. b) Proportion of candidates tested as full constructs, partial constructs or both. c) Barplot showing recombinant protein system used for each protein based tested antigen. d) Bar and dotplot showing adjuvants used for each tested antigen, with adjuvant characteristics indicated on the dotplot below. e) Barplot showing animal model used for each tested antigen. f) Barplot showing mouse strain used for each antigen tested in a mouse model. g) Barplot showing the number of immunisation doses used for each tested antigen. h) Barplot showing administration route for each vaccine antigen. i) Barplot showing cercarial challenge dose for each vaccine antigen. j) Barplot showing gap between final immunisation and challenge for each vaccine antigen. k) Barplot showing gap between cercariae challenge and cull (measurement of worm/egg counts) for each vaccine antigen.
(TIF)

**S1 Table. Underlying data.** Database of pre-clinical vaccine efficacy testing underlying analyses.
(XLSX)

## Acknowledgments

Thank you to Floor de Weijer for her expertise in creating the data portal and Shiny app.

## Author contributions

**Conceptualization:** Emma Louise Houlder, Angela van Diepen, Murilo Sena Amaral, R. Alan Wilson, Cornelis H. Hokke, Meta Roestenberg, Wilfried A.M. Bakker.

**Data curation:** Emma Louise Houlder.

**Formal analysis:** Emma Louise Houlder, Lucas Ferreira da Silva.

**Funding acquisition:** Emma Louise Houlder, Meta Roestenberg, Wilfried A.M. Bakker.

**Investigation:** Emma Louise Houlder, Lucas Ferreira da Silva.

**Methodology:** Emma Louise Houlder, Lucas Ferreira da Silva.

**Project administration:** Emma Louise Houlder, Meta Roestenberg.

**Software:** Emma Louise Houlder, Lucas Ferreira da Silva.

**Supervision:** Angela van Diepen, Murilo Sena Amaral, R. Alan Wilson, Cornelis H. Hokke, Meta Roestenberg, Wilfried A.M. Bakker.

**Visualization:** Emma Louise Houlder, Lucas Ferreira da Silva.

**Writing – original draft:** Emma Louise Houlder.

**Writing – review & editing:** Emma Louise Houlder, Lucas Ferreira da Silva, Angela van Diepen, Murilo Sena Amaral, R. Alan Wilson, Cornelis H. Hokke, Meta Roestenberg, Wilfried A.M. Bakker.

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
