## [Decision Letter · Decision Letter 0]

11 Apr 2025

*Schistosoma mansoni*
Response to Reviewers
Revised Manuscript with Track Changes
Manuscript

Shaden Kamhawi

co-Editor-in-Chief

Paul Brindley

co-Editor-in-Chief

**Journal Requirements:**

At this stage, the following Authors/Authors require contributions: Emma Louise Houlder, Lucas Ferreira da Silva, Angela van Diepen, Murilo Sena Amaral, R. Alan Wilson, Cornelis H. Hokke, Meta Roestenberg, and Wilfried A.M. Bakker. Please ensure that the full contributions of each author are acknowledged in the "Add/Edit/Remove Authors" section of our submission form.

3) Please amend your detailed Financial Disclosure statement. This is published with the article. It must therefore be completed in full sentences and contain the exact wording you wish to be published. Please ensure that the funders and grant numbers match between the Financial Disclosure field and the Funding Information tab in your submission form. Note that the funders must be provided in the same order in both places as well.

**Reviewers' comments:**

**Key Review Criteria Required for Acceptance?**

**Methods:**

-Are the objectives of the study clearly articulated with a clear testable hypothesis stated?

-Is the study design appropriate to address the stated objectives?

-Is the population clearly described and appropriate for the hypothesis being tested?

-Is the sample size sufficient to ensure adequate power to address the hypothesis being tested?

-Were correct statistical analysis used to support conclusions?

-Are there concerns about ethical or regulatory requirements being met?

Reviewer #1: The review methods and statistics are appropriate

Reviewer #2: This paper is comprehensive and provides significant added value to the literature. Many reviews of Schistosoma vaccines exist, but none systematically compare different vaccines and vaccine parameters. This work will be invaluable for investigators designing pre-clinical vaccine trials and selecting antigens to push into clinical trials.

The review's objectives are articulated and provide a comprehensive overview of the current landscape in schistosomiasis vaccine development.

**Results:**

-Does the analysis presented match the analysis plan?

-Are the results clearly and completely presented?

-Are the figures (Tables, Images) of sufficient quality for clarity?

Reviewer #1: The analysis resulted in excellent recommendations

Reviewer #2: The authors have clearly outlined a systematic and organized approach to reviewing preclinical data, and they remain consistent with that framework throughout the manuscript.

The table and figures are easy to interpret and contribute meaningfully to understanding the manuscript.

It would be helpful to note which vaccines have been tested with adjuvants/platforms that are approved for human use (or have a corollary like AddaVax/MF59). It is very relevant to the translatability of a vaccine candidate if its maximum efficacy is achieved with a system usable in humans versus one that is not.

Another variable that could be considered is whether any candidates have been tested independently by multiple research groups.

**Conclusions:**

-Are the conclusions supported by the data presented?

-Are the limitations of analysis clearly described?

-Do the authors discuss how these data can be helpful to advance our understanding of the topic under study?

-Is public health relevance addressed?

Reviewer #1: The conclusions are relevant as the present the state of the field

Reviewer #2: They accurately synthesize the findings and highlight key trends in schistosomiasis vaccine research, including promising candidates and recurring challenges. The authors strike a good balance between optimism and realism, reflecting the field's current state.

**Editorial and Data Presentation Modifications?**

Reviewer #1: Line 78; worm and/or egg counts. Recommend that both worm burden reduction and egg count be recommended. This reviewer realizes that egg count is a function of worm burden.

Reviewer #2: Edits/minor changes

• Line 38: italicize Schistosoma

• Line 43: comma after “design”

• Sometimes there is a space between the end of the sentence and the parenthetical citation, sometimes there is not.

• Line 66: could be more clear to say “protection against schistosome infection”

• Line 194: comma after “systems”

• Line 195: comma after “efficacious”

• Line 199-201: comma after “this” and/or could be reworded to be easier to follow

• Figure 4a: hamsters are mentioned in the text but not on the graph

• Figures in general: it is not entirely clear to me what “count per candidate” refers to – is this the number of tests, the number of papers, etc.?

• Line 239: comma after “route”

• Line 281: delete extra parenthesis

• Line 294: comma after “parasite”

• Line 413: add “are” between “post-vaccination” and “crucial”

**Summary and General Comments:**

Reviewer #1: The goal of this review is to present an overview of pre-clinical prophylactic vaccine studies targeting S. mansoni in the last 30 years (1994-2024). The article summarizes what is known about the antigens tested, their overall efficacy, the adjuvant and platform used, and the timing of immunization and challenge. Importantly the authors evaluate how the choice of these parameters may influence the efficacy reached in preclinical trials and discuss how these parameters translate to clinical vaccination trials.

Of the141 vaccines tested that met the inclusion criteria only 10 achieved an efficacy of 60% and of the 10 only 2 reached an efficacy of >90% in pre-clinical studies. The outcome of the review of these vaccine preclinical trials is that to compare trials, researchers must consider employing consistent and optimized vaccine formulation and study design parameters.

One of the strengths of the review is that the authors have summarized in Tables the formulations and study designs used, and highlighted how certain parameters affect efficacy and with that information have made recommendations to researchers on how to perform vaccine tests in the future. This information will be useful for established labs to rethink their approach but also for new investigators.

A weakness is that the study focuses only on S. mansoni and the mouse model. Never-the- less, it is useful to point out the state of the vaccine field.

The outcome of the study is that development of a vaccine for schistosomiasis is not likely in the near future unless laboratories decide on consistent and comparable approaches. However, innovation is strongly encouraged.

Reviewer #2: This well-written and timely review offers a comprehensive and insightful synthesis of current progress in developing schistosomiasis vaccines. The manuscript is well-structured and logically organized and reflects a thorough command of the existing literature. It effectively highlights key advances in preclinical studies and provides a thoughtful analysis of the challenges.

PLOS authors have the option to publish the peer review history of their article (what does this mean? ). If published, this will include your full peer review and any attached files.

**Do you want your identity to be public for this peer review?** For information about this choice, including consent withdrawal, please see our Privacy Policy .

Reviewer #1: No

Reviewer #2: No

**Figure resubmission:****Reproducibility:** To enhance the reproducibility of your results, we recommend that authors of applicable studies deposit laboratory protocols in protocols.io, where a protocol can be assigned its own identifier (DOI) such that it can be cited independently in the future. Additionally, PLOS ONE offers an option to publish peer-reviewed clinical study protocols. Read more information on sharing protocols at https://plos.org/protocols?utm_medium=editorial-email&utm_source=authorletters&utm_campaign=protocols

---

## [Editor Report · Decision Letter 1]

13 May 2025

Dear Dr Houlder,

We are pleased to inform you that your manuscript 'Pre-clinical studies of *Schistosoma mansoni* vaccines: a scoping review' has been provisionally accepted for publication in PLOS Neglected Tropical Diseases.

Best regards,

David J. Diemert, M.D.

Academic Editor

Jong-Yil Chai

Section Editor

Shaden Kamhawi

co-Editor-in-Chief

Paul Brindley

co-Editor-in-Chief

---

## [Editor Report · Acceptance letter]

Dear Dr Houlder,

We are delighted to inform you that your manuscript, "Pre-clinical studies of *Schistosoma mansoni* vaccines: a scoping review," has been formally accepted for publication in PLOS Neglected Tropical Diseases.

Best regards,

Shaden Kamhawi

co-Editor-in-Chief

Paul Brindley

co-Editor-in-Chief
